# Diagnostic Value of the Combined Measurement of Serum HCY and NRG4 in Type 2 Diabetes Mellitus with Early Complicating Diabetic Nephropathy

**DOI:** 10.3390/jpm13030556

**Published:** 2023-03-20

**Authors:** Sheng Ding, Yi Yang, Yuming Zheng, Jinling Xu, Yangyang Cheng, Wei Wei, Fuding Yu, Li Li, Menglan Li, Mengjie Wang, Zhongjing Wang, Guangda Xiang

**Affiliations:** 1The First School of Clinical Medicine, Southern Medical University, Guangzhou 510515, China; 2Department of Endocrinology, General Hospital of Central Theater Command, Wuluo Road 627, Wuhan 430070, China; 3Department of Endocrinology, The Central Hospital of Wuhan, Tongji Medical College, Huazhong University of Science and Technology, Wuhan 430014, China; 4Department of Radiology, The First Wuhan Hospital, Wuhan 430022, China; 5Department of Physical Examination, The Central Hospital of Wuhan, Tongji Medical College, Huazhong University of Science and Technology, Wuhan 430014, China

**Keywords:** HCY, NRG4, HCY/NRG4, diabetes, DKD

## Abstract

Purpose: This study aimed to investigate the value of combined detection of HCY and NRG4 in the diagnosis of early diabetic kidney disease (DKD) and to explore the association between the ratio of HCY/NRG4 and DKD. Methods: A total of 140 diabetic patients and 43 healthy people were prospectively enrolled. The plasma HCY level, NRG4 level and HCY/NRG4 of them were measured to compare their differences and analyze the correlation with DKD. The independent influencing factors of patients with DKD were screened, and the nomograph of DKD occurrence was constructed. Results: The levels of HCY and HCY/NRG4 in diabetic patients were significantly increased, while the level of NRG4 was significantly decreased (*p* < 0.01). The AUCs of HCY/NRG4 predicted for DKD were 0.961. HCY/NRG4 and the course of DM were independent risk factors for DKD. A predictive nomograph of DKD was constructed, and decision curve analysis (DCA) showed good clinical application value. HCY/NRG4 was positively correlated with Scr, UACR, TG, UA, BUN, TCHOL and LDL and negatively correlated with eGFR and HDL (*p* < 0.05). Conclusions: The level of HCY and NRG4 is closely related to the severity of DM, and combined detection of HCY/NRG4 can identify patients with DKD at an early stage.

## 1. Introduction

According to the statistical report of the World Health Organization, in the past 34 years, the number of people living with diabetes has reached 314 million, and due to the increasing incidence and mortality of diabetes, this disease is expected to become the seventh leading cause of death by 2030 [1]. There are many types of diabetes, of which type 2 diabetes mellitus (T2DM) is an increasing health problem worldwide due to the body’s inability to produce insulin or use insulin from the pancreas [2], and insulin resistance is a major underlying pathophysiology of T2DM. T2DM is a chronic, noninfectious, multisystem disease in which chronic exposure to hyperglycemia affects the microvascular and macrovascular systems, leading to diabetic nephropathy (DN), retinopathy, neuropathy and cardiovascular disease, with significant implications for quality of life and overall life expectancy [3,4]. The main characteristics of DN are persistent proteinuria, elevated creatinine levels and decreased glomerular filtration rate. Pathologically, podocyte injury, glomerulosclerosis, glomerular basement membrane thickening, interstitial fibrosis and tubular atrophy mainly contribute to the development of DN [5]. Podocyte insulin resistance is a main cause of podocyte injury, playing a crucial role in contributing to albuminuria in early DN [6]. DN is an important cause of increased morbidity and mortality in diabetic patients [7]. A meta-analysis found that the prevalence of DN was relatively high in Chinese patients with T2DM, with a total co-morbidity of 21.8%, and there were regional and gender differences. These data indicated that the risk and burden of DN to society increased year by year [8].

DN is the main cause of end-stage renal disease, and strict management of modifiable risk factors is essential to prevent and delay renal decline. The new markers may help in the early diagnosis of this common and serious complication, but further research is needed to clarify their effectiveness [9]. Studies have shown that DN is a chronic inflammatory disease mediated by a range of cytokines, including interleukin-6, interleukin-8, homocysteine (HCY) and cystatin C (Cys C). Among them, HCY is believed to be related to microvascular complications in diabetic patients [9,10,11]. Studies have found that HCY is closely related to the occurrence and development of DN. Wang et al. [12] showed that serum HCY and Cys C levels were consistent with the occurrence and progression of DN, and serum HCY and Cys C were sensitive biomarkers for detecting early DN and monitoring its progression. Ye et al. [13] found that serum HCY level in the DN group was higher than that in the T2DM group, which was associated with kidney damage, and could be used as a potential serological indicator for the early diagnosis of DN. In addition, a meta-analysis suggested that serum HCY was a promising biomarker of DN [14]. Therefore, HCY may be involved in the occurrence and progression of DN, which can be used for the early prediction of DN.

Neuregulin 4 (NRG4) is a novel adipokine released from brown adipose tissue. It plays a key role in regulating overall body energy balance, glycolipid metabolism and reducing chronic inflammation [15,16]. It has been reported that decreased NRG4 levels are closely associated with type 2 diabetes, obesity, hyperglycemia, oxidative stress and inflammation. Yan et al. [17] measured plasma NRG4 levels in T2DM patients by ELISA and found that the level of NRG4 was negatively correlated with most of the metabolic syndrome analysis (MetS) indicators, and the decrease of plasma NRG4 level may be related to oxidative stress, inflammation and dyslipidemia, which may be related to the occurrence of MetS in T2DM patients. A cross-sectional study of T2DM patients without diabetic peripheral neuropathy (DPN) showed significantly lower levels of circulating NRG4 than that of the control group, and the level of circulating NRG4 was further reduced in T2DM patients with DPN. The level of circulating NRG4 decreased gradually with the increase in screening for abnormal DPN. NRG4 may be a novel adipokine associated with inflammation, oxidative stress and long-term glycemic control in patients with T2DM [18]. Kocak et al. [19] found that NRG4 decreased 1.9 times in patients with diabetic microvascular complications compared with the normal group. In patients with diabetes, NRG4 levels may be a good predictor of early detection of one or more diabetic microvascular complications. The above studies indicate that NRG4 may be a potential predictor of DN.

However, at present, to improve the early prediction and detection level of diabetic nephropathy, it is still necessary to further study new non-invasive diagnostic markers. The prediction of a single indicator often has limitations, and feasible measures for diagnosing DN before advanced renal insufficiency are considered to be of clinical significance. Therefore, the purpose of this study was to evaluate serum HCY and NRG4 levels in T2DM patients with DN and to explore the predictive value of serum HCY combined with NRG4 levels in the early detection of type 2 diabetic nephropathy.

## 2. Materials and Methods

### 2.1. Selection of Patients and the Research Design

Inclusion criteria: ① Patients with type 2 diabetes; ② Han population; ③ Age over 18 years old; ④ Body mass index (BMI) is between 18.5–30 kg/m^2^. Exclusion criteria: ① Patients with type 1 diabetes or other special types of diabetes; ② Non-diabetes nephropathy patients with chronic kidney disease; ③ Patients with glomerular filtration rate (eGFR) less than 10 mL/min/1.73 m^2^ requiring dialysis; ④ Complicated with acute complications of diabetes, such as diabetic ketoacidosis and diabetic lactic acidosis; ⑤ Recently, cerebral infarction, acute myocardial infarction and peripheral vascular obstructive disease were combined; ⑥ In recent 6 months, he has used drugs to delay the development of diabetic nephropathy, such as DPP4 inhibitor, GLP-1 receptor agonist, SGLT-2 inhibitor hypoglycemic drugs, ACEI, ARBs antihypertensive drugs; ⑦ Complicated with acute infection, tumor or liver disease; ⑧ Pregnancy status. At the same time, healthy subjects matched with age were selected as the control group. A total of 140 diabetes patients in the Central Hospital of Wuhan were prospectively enrolled, including 55 type 2 diabetes patients (DM group) and 85 diabetic kidney disease patients (DKD group); 43 healthy people (NC group) in the same period were selected as the control group. This study was approved by the Medical Ethics Committee of the Central Hospital of Wuhan (NO.2021(11)-01), and all subjects signed the informed consent form.

### 2.2. Data Collection and Definition

The subjects’ basic information (gender, age, medical history, medication history, BMI, blood pressure, waist circumference, etc.) was recorded; urinary albumin/creatinine ratio (UACR), blood lipids including Low-Density Lipoprotein (LDL), cholesterol (TCHOL), High-Density Lipoprotein (HDL), triglyceride (TG), serum creatinine (Scr), blood urea nitrogen (BUN), blood uric acid (UA), blood glucose, glycosylated hemoglobin (HbA1C), glomerular filtration rate (eGFR) and inflammatory factors were determined. Insulin or C-peptide (calculation of HOMA-IR and HOMA-β), plasma HCY and NRG4 levels were measured by ELISA. DM or DKD is defined as meeting the 2018 ADA diagnostic criteria for type 2 diabetes and diabetic kidney disease [20]. Albuminuria was defined as UACR greater than 30 mg/g on 2 or more consecutive 3 urine tests in the last 6 months.

### 2.3. Serum HCY and Neuregulin-4 Measurement

Serum HCY (AUSA Co., Ltd., Shenzhen, China) and NRG4 (Zell Bio GmbH, Lonsee, Germany) were measured using an enzyme-linked immunosorbent assay (ELISA) according to the provided instructions. The sensitivity of the kit was 4 μmol/L and 0.02 ng/mL, respectively. The standard curve was developed by a linear range of the standard for each cytokine and used for the calculation of the concentrations. The intra- and inter-assay variations were <10%.

### 2.4. Statistical Analysis

SAS9.4 (SAS Institute Inc., Cary, NC, USA) software was used for statistical analysis. The measurement data were described by mean ± standard deviation. Two independent samples t-test was used for comparison between the two groups, and one-way ANOVA was used for comparison between multiple groups. Counting data were expressed by examples (%) and compared between groups by χ^2^ inspection. Logistic regression was used to analyze the relationship between clinical characteristics, laboratory test results and DKD. Then, analyze and evaluate the predictive ability of HCY/NRG4 to patients’ DKD by drawing the ROC. To further test the accuracy of the histogram in predicting disease occurrence, a calibration curve was generated, and the observations were compared with the predictions. In addition, Pearson correlation was used to analyze the correlation between laboratory test indicators and HCY/NRG4. *p* < 0.05 was considered statistically significant.

## 3. Results

### 3.1. Clinical Data Characteristics of Patients

A total of 140 patients with diabetes were prospectively included in this study; 55 patients with diabetes, 85 patients with DKD and 43 healthy controls were selected at the same time. The UACR, HCY, HCY/NRG4, FBG, HbA1C, BUN, Scr, LDL, TG and UA levels in DM and DKD groups were higher than those in the NC group (*p* < 0.05 or *p* < 0.01). NRG4 and eGFR were lower than those in the NC group (*p* < 0.05 or *p* < 0.01) (Table 1). Compared with the healthy control group, the level of HCY and HCY/NRG4 in DM and DKD groups were significantly increased, while the level of NRG4 was significantly decreased, with statistical significance (*p* < 0.01) (Figure 1A–C).

### 3.2. The Predictive Efficacy of HCY, NRG4 and HCY/NRG4 in Predicting DKD

The prediction specificity and sensitivity of NRG4 for DKD were 0.745 and 0.941, and the prediction specificity and sensitivity of HCY for DKD were 0.909 and 0.824. The specificity and sensitivity of HCY/NRG4 for predicting DKD were 0.927 and 0.929. The AUCs predicted for DKD were NRG4 (0.91, 95%CI: 0.859, 0.961), HCY (0.885, 95%CI: 0.822, 0.948), and HCY/NRG4 (0.961, 95%CI: 0.928, 0.994), respectively (Figure 1D).

### 3.3. Analysis of Multiple Factors Affecting DKD Occurrence

Multivariate logistic regression analysis was performed on all the factors affecting the occurrence of DKD in patients, and the results showed that HCY/NRG4 (HR: 1.870, CI: 1.496–2.573, *p* < 0.001); course of DM (HR: 1.015, CI: 1.004–1.029, *p* = 0.012) was an independent factor influencing the occurrence of DKD in patients (Table 2).

### 3.4. Construction and Clinical Value of Predictive Nomogram

Based on the multivariable logistic regression results of DKD, we finally selected HCY/NRG4 and the course of DM as two valuable factors to establish the prediction model (Figure 2A). In our cohort, the calibration curve of predicting patients’ DKD nomogram shows that there is good consistency between prediction and observation, and the calibration curve of the nomogram has no deviation from the reference line, with high reliability (Figure 2B). Decision curve analysis (DCA) is a novel strategy for evaluating alternative predictive treatment methods and has advantages over the Area Under the Receiver Operating Characteristic Curve (AUROC) in clinical value evaluation. The DCA curves for the developed nomogram in the cohorts are presented in Figure 2C. The DCA of the nomogram has higher net benefits, indicating that it had better clinical outcome values.

### 3.5. Pearson Correlation Analysis between Serum HCY/NRG4 Level and Other Indicators

Pearson correlation analysis showed that HCY/NRG4 was positively correlated with Scr, UACR, TG, UA, BUN, TCHOL and LDL (*p* < 0.05), negatively correlated with eGFR and HDL (*p* < 0.05 or *p* < 0.01) (Figure 3).

## 4. Discussion

A total of 140 diabetic patients were included in this study. Compared with the healthy control group, the levels of HCY and HCY/NRG4 in DM and DKD groups were significantly increased, while the levels of NRG4 were significantly decreased. The predictive specificity and sensitivity of NRG4 for DKD were 0.745 and 0.941, and that of HCY for DKD were 0.909 and 0.824, respectively. The specificity and sensitivity of DKD predicted by HCY/NRG4 were 0.927 and 0.929, respectively. Logistic regression analysis of all factors influencing the development of DKD in patients showed that HCY/NRG4 and the course of DM were independent factors influencing the development of DKD in patients. These two valuable factors were selected to establish a nomogram, and the results of DCA showed that the nomogram had better clinical prediction value.

Early prediction of T2DM with diabetic nephropathy has also been reported. Shoukry et al. [21] found that urinary monocyte chemotactic protein-1 (MCP-1) and vitamin D-binding protein(VDBP) levels in diabetic patients were significantly higher than those in the normal group, and the ROC curve analysis of urinary MCP-1 and urinary VDBP levels showed high sensitivity and specificity for the early diagnosis and detection of DN. The optimal cut-off point of uMCP-1 for predicting DN was 110 pg/mg (AUC = 0.987). However, the optimal cut-off point of uVDBP for predicting DN was 550 ng/mg (AUC = 0.947). Urinary MCP-1 and urinary VDBP levels may be considered as novel potential diagnostic biomarkers for the early detection of diabetic nephropathy. Lee et al. [22], in a clinically based cross-sectional study of 320 patients with type 2 diabetes who underwent staging of diabetic nephropathy and evaluated the prognosis of type 2 diabetic nephropathy based on serum creatinine and cystatin C (CysC), found that serum CysC seemed to predict prognosis more accurately than serum creatinine; CysC-based GFR may be more valuable than creatinine-based GFR in predicting the stage of microalbuminuria. A meta-analysis found that CysC predicted DN with sensitivity and specificity of 0.88 and 0.85, the positive predictive value of DN was 7.04, and the area under the ROC curve was 0.9549, which could be considered an early predictor of DN [23]. The AUCs predicted for DKD were HCY/NRG4 (0.961), CysC (0.9549), urinary MCP-1 (0.987) and VDBP (0.947), showing early diagnostic value for DN of these biomarkers. Inflammatory biomarkers, such as TNF-α and IL-1β, also play a predictive role in DN [24]. Previous studies have confirmed that the levels of IL-1β and TNF-α produced by macrophages cultured in the glomerular basement membrane of diabetic rats are significantly higher than those produced by macrophages cultured in the basement membrane of normal non-diabetic rats, indicating that these pro-inflammatory cytokines can be involved in the development of DN [25]. Clinical studies have shown a direct and significant relationship between urinary protein excretion and serum TNF-α in patients with normal renal function and diabetes, and the fact that urinary TNF-α excretion increases significantly with DN progression strongly supports the prospect of using this cytokine as a biomarker for predicting DN [26,27]. Although these biomarkers have a role to play in the assessment of diabetic nephropathy, the current data still rule out most biomarkers for routine clinical use. However, the trajectory of research on novel biomarkers of DN should be a sustained effort to validate them through high-quality and large-scale longitudinal studies and subsequently develop DN biomarkers capable of reliably predicting and evaluating them [28].

In recent years, serum HCY and NRG4 have increased in studies on T2DM and DN, which may be potential biomarkers for the early prediction of DN [29]. In a study on the risk of circulating homocysteine and DKD in the population, it was found that for every 5 μmol/L increase in blood homocysteine concentration, the odds ratio of DKD to diabetes was 3.86. In logistic regression analysis, hypertension, homocysteine and triglycerides were significantly associated with an increased risk of DKD. It is suggested that there is a causal relationship between increased circulating homocysteine concentration and increased risk of DKD [30]. In addition, recent data show that the expression of NRG4 is significantly down-regulated in mice and human obesity. NRG4 may enhance the activity of brown adipose tissue, increase the expression of thermogenic markers, reduce the expression of lipogenic/adipogenic genes, aggravate the browning of white adipose tissue, promote the oxidation and ketogenesis of liver fat, induce neurite growth and enhance the blood vessels of adipose tissue, to prevent obesity and related metabolic complications [31]. Ye et al. [13] found that the sensitivity, specificity and AUC of serum HCY level in the diagnosis of DN were 84.31%, 74.55% and 0.85, showing high sensitivity, specificity and AUC. In our study, HCY showed higher specificity and sensitivity in predicting DKD with more data samples. A prospective observational study similar to our findings found that baseline levels of HCY in patients with DN were significantly elevated and correlated with disease severity, supporting plasma HCY as an independent risk factor for DN and an early predictor of DN progression in patients with type 2 diabetes [32]. Some studies have found that the level of circulating NRG4 in patients with metabolic syndrome is lower than that in the healthy control group, and the concentration of NRG4 is negatively correlated with the risk of developing metabolic syndrome, and NRG4 concentration may be a protective factor [33]. However, studies have found no correlation between the circulating NRG4 level and the prevalence of diabetic nephropathy and diabetic retinopathy [34]. In our study, through the analysis of large sample data, it was found that NRG4 was closely related to the incidence of HCY and DN, and the combination of NRG and HCY had excellent predictive efficacy in predicting early DN of T2DM.

However, this study still has some limitations. First, our enrolled samples are not enough, and we lack an internal validation cohort for verification. Second, as this study is a single-center study, further multi-center prospective clinical studies are needed to prove the clinical validity of this model. Third, only clinicopathological features were included in the variable analysis, and molecular pathologic features should be included to further improve the nomogram prediction. However, our study is also very important clinically. We used the serum HCY/NRG4 ratio to predict T2DM with early DN, and the AUC curve area reached 0.96, suggesting that this new indicator can be considered as an important factor of T2DM with early DN. Our study developed an auxiliary model to predict the onset of T2DM with early DN, but this auxiliary model should be used with caution based on the overall situation of the patient.

## 5. Conclusions

In summary, the levels of HCY and NRG4 were closely related to the severity of DKD in T2DM patients with early DKD. Combined HCY/NRG4 detection can detect the occurrence of DKD in diabetic patients at an early stage. Further study of the mechanism of how NRG4 plays a protective role in patients with diabetic nephropathy would be necessary for the future.

## Figures and Tables

**Figure 1 jpm-13-00556-f001:**
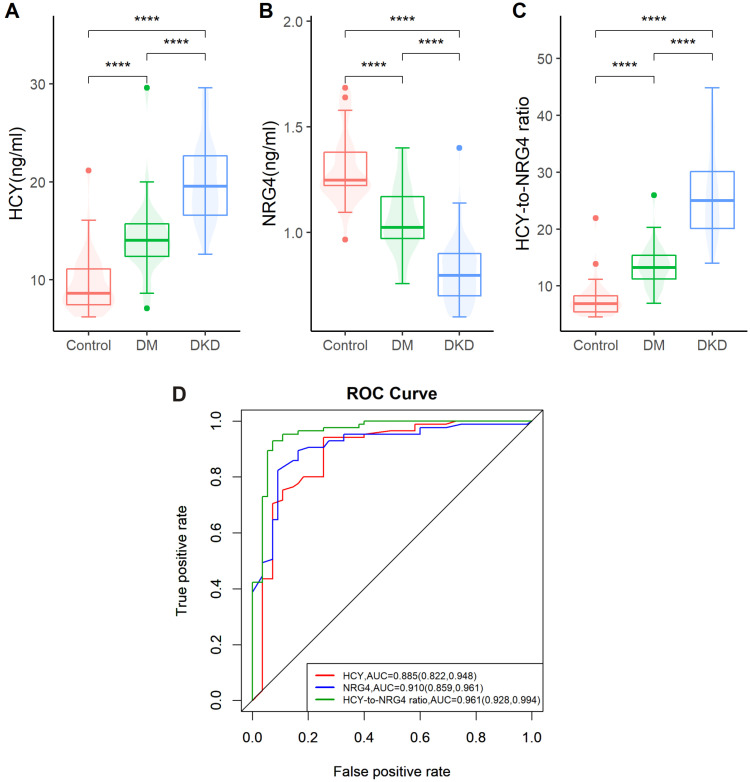
Comparison of NRG4, HCY and HCY/NRG4 between groups and their predictive efficacy. (**A**) Comparison of HCY among the control group, the DM group and the DKDs; (**B**) Comparison of NRG4 among the control group, the DM group and the DKDs; (**C**) Comparison of HCY/NRG4 among the control group, the DM group and the DKDs; (**D**) The predictive efficacy of NRG4, HCY and HCY/NRG4 in DKD. **** *p* < 0.01.

**Figure 2 jpm-13-00556-f002:**
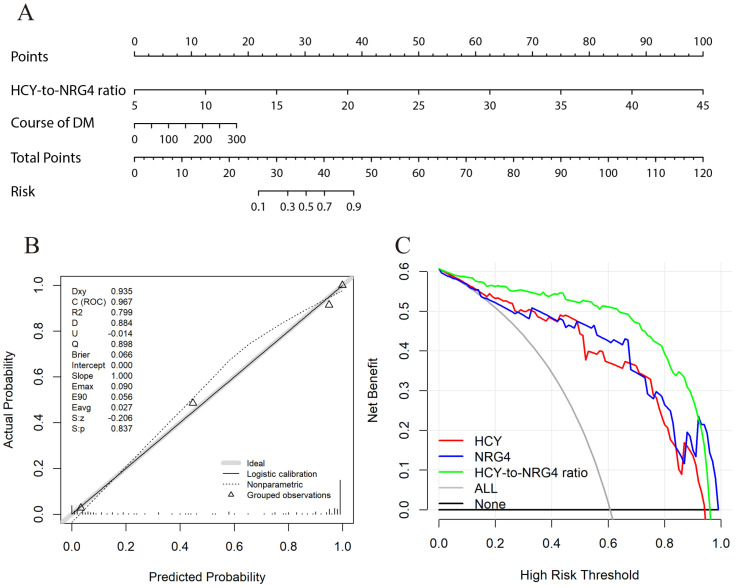
Evaluation of DKD rates associated nomograms, the calibration curves and decision curve analysis for patients with DM. (**A**) The nomogram integrating the HCY/NRG4 and course of DM for predicting the risk of DKD; (**B**) The calibration curve for predicting patients’ risk of DKD; (**C**) Decision curve analysis of the nomogram for the risk of DKD.

**Figure 3 jpm-13-00556-f003:**
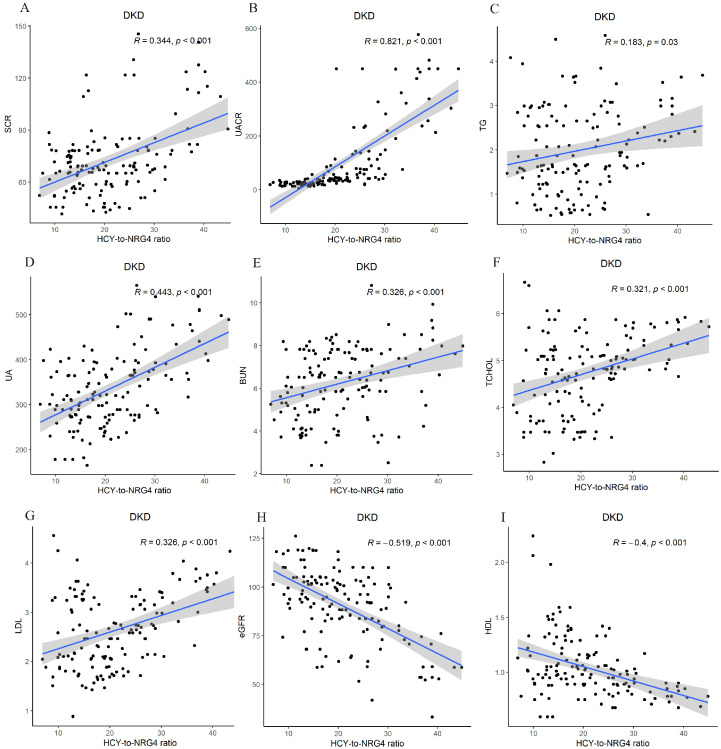
Pearson correlation analysis between serum HCY/NRG4 level and other indicators. Scatter plots showing the correlation between HCY/NRG and SCR, UACR, TG, UA, BUN, TCHOL, LDL, GFR and HDL in patients with DKD. HCY/NRG4 was positively correlated with SCR, UACR, TG, UA, BUN, TCHOL and LDL (**A**–**G**) but negatively correlated with GFR and HDL (**H**,**I**).

**Table 1 jpm-13-00556-t001:** Comparison of baseline data and laboratory test indexes among the three groups.

Characteristics		Control (N = 43)	DM (N = 55)	DKD (N = 85)
Gender (%)	Male	16 (37.21)	19 (34.55)	41 (48.24)
	Female	27 (62.79)	36 (65.45)	44 (51.76)
Age (mean (SD)), y		45.49 (18.72)	53.02 (16.09) a	55.27 (15.41) b
BMI (mean (SD)), kg/m^2^		22.29 (2.89)	23.72 (4.00) a	23.71 (3.33) b
UACR (mean (SD)), mg/g		15.66 (6.83)	21.54 (5.85) a	147.75 (147.54) b,c
eGFR (mean (SD)), ml/min		103.38 (15.30)	96.75 (17.50)	85.74 (18.66) b,c
HCY (mean (SD)), μmol/l		9.46 (2.84)	14.34 (3.96) a	20.16 (4.08) b,c
NRG4 (mean (SD)), ng/ml		1.31 (0.17)	1.07 (0.16) a	0.80 (0.14) b,c
HCY-to NRG4 ratio (mean (SD))		7.44 (2.99)	13.61 (3.73) a	25.98 (7.30) b,c
FBG (mean (SD)), mmol/l		5.61 (1.08)	9.73 (4.71) a	10.03 (4.34) b
HbA1C (mean (SD))		5.34 (0.41)	9.12 (1.63) a	9.33 (1.72) b
BUN (mean (SD)), mmol/l		4.33 (0.94)	5.60 (1.44) a	6.70 (1.65) b,c
SCR (mean (SD)), μmol/l		65.70 (12.16)	70.13 (17.37)	74.27 (23.14) b
LDL (mean (SD)), mmol/l		1.89 (0.71)	2.61 (0.81) a	2.65 (0.70) b
TCHOL (mean (SD)), mmol/l		4.14 (0.82)	4.59 (0.95) a	4.83 (0.71) b
HDL (mean (SD)), mmol/l		1.54 (0.58)	1.11 (0.34) a	0.99 (0.22) b,c
TG (mean (SD)), mmol/l		1.41 (0.95)	1.90 (0.84)	1.97 (0.96) b
UA (mean (SD)), μmol/l		298.00 (278.50)	307.80 (65.66)	354.66 (90.43) b,c
Course of DM (mean (SD)), month			48.16 (65.89)	109.17 (87.31) c
DR (%)	No		41 (74.55)	41 (48.24) c
	Yes		14 (25.45)	44 (51.76)
PAD (%)	No		21 (38.18)	30 (35.29)
	Yes		34 (61.82)	55 (64.71)
DPN (%)	No		25 (45.45)	38 (44.71)
	Yes		30 (54.55)	47 (55.29)
HBP (%)	No		36 (65.45)	41 (48.24)
	Yes		19 (34.55)	44 (51.76)
CHD (%)	No		27 (49.09)	52 (61.18)
	Yes		28 (50.91)	33 (38.82)
Stroke (%)	No		38 (69.09)	55 (64.71)
	Yes		17 (30.91)	30 (35.29)

a: control vs. DM; b: control vs. DKD; c: DM vs. DKD; BMI: Body mass index; UACR: Urinary albumin/creatinine ratio; eGFR: Glomerular filtration rate; HCY: Homocysteine; NRG4: Neuregulin 4; FBG: Fast blood glucose; HbA1C: Glycosylated Hemoglobin; BUN: Blood urea nitrogen; SCR: Serum creatinine; LDL: Low-density lipoprotein; TCHOL: Total cholesterol; HDL: High-density lipoprotein; TG: Triglyceride; UA: Uric acid; DR: Diabetic retinopathy; PAD: Peripheral vascular disease; DPN: Diabetic peripheral neuropathy; HBP: High blood pressure; CHD: Coronary heart disease.

**Table 2 jpm-13-00556-t002:** Multivariable logistic regression analyses for identifying factors independently associated with DKD patients.

	OR	95% CI		*p*
		Lower Limit	Upper Limit	
HCY-to-NRG4 ratio	1.870	1.496	2.573	<0.001 *
BUN	1.441	0.917	2.356	0.121
HDL	1.094	0.063	17.802	0.949
UA	0.993	0.983	1.004	0.227
DR	0.432	0.391	7.510	0.323
Course of DM	1.015	1.004	1.029	0.012 *

HCY/NRG4: HCY-to-NRG4 ratio; BUN: Blood Urea Nitrogen; HDL: High-Density Lipoprotein; UA: Uric Acid; DR: Diabetic Retinopathy. * *p* < 0.05

## Data Availability

Not applicable.

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
