# Peer review of "Diagnostic Value of the Combined Measurement of Serum HCY and NRG4 in Type 2 Diabetes Mellitus with Early Complicating Diabetic Nephropathy"

_jpm, 2023, doi:10.3390/jpm13030556_

Round 1

Reviewer 1 Report

Manuscript ID: jpm-2260832 Dear authors,   I congratulate your for conducting an interesting study on the association between the ratio of Serum HCY /NRG4 and DKD.   Here are my comments:   1. Table 1: include full form of abbreviations for all the table contents   2. Likewise, in other places VDBP, MCP-1

3. What about the measurement of postprandial blood glucose levels?   4. How significant is the combined HCY/NRG4 detection than the other mentioned biomarkers like CysC,urinary MCP-1 and VDBP   5. Add the main symptoms of diabetic nephropathy in the introduction.   6. Add the feature research perspectives in the conclusion section

Author Response

Point 1: Table 1: include full form of abbreviations for all the table contents.

Response 1: We are truly grateful for your valuable comments and suggestions. We have added the full form of abbreviations for all the table contents.

Point 2: Likewise, in other places VDBP, MCP-1

Response 2: Thank you very much for the comments. We have modified the full form of abbreviations for VDBP, MCP-1 in line 228.

Point 3: What about the measurement of postprandial blood glucose levels?

Response 3: It’s a great idea to measure postprandial blood glucose levels. We didn`t show the postprandial blood glucose levels due to unstandardized food intake.

Point 4: How significant is the combined HCY/NRG4 detection than the other mentioned biomarkers like CysC,urinary MCP-1 and VDBP.

Response 4: We are truly grateful for your valuable comments. The AUCs predicted for DKD were HCY/NRG4 (0.961), CysC (0.9549), urinary MCP-1(0.987) and VDBP(0.947) showing early diagnostic value for DN of these biomarkers. The optimal cut-off point of uMCP-1 for predicting DN was 110 pg/mg (AUC = 0.987). However, the optimal cut-off point of uVDBP for predicting DN was 550 ng/mg (AUC = 0.947). We have added these sentences in discussion.

Point 5: Add the main symptoms of diabetic nephropathy in the introduction.

Response 5: Thank you very much for the comments. We strongly agree with you. We have added the following sentence in the introduction. The main characteristics of DKD are persistent proteinuria, elevated creatinine levels and decreased glomerular filtration rate. Pathologically, podocyte injury, glomerulosclerosis, glomerular basement membrane thickening, interstitial fibrosis, and tubular atrophy mainly contribute to the development of DKD.

Reference:Jiang A, Song A, Zhang C. Modes of podocyte death in diabetic kidney disease: an update. J Nephrol. 2022;35(6):1571-1584. doi:10.1007/s40620-022-01269-1

Point 6: Add the feature research perspectives in the conclusion section

Response 6: It’s a great idea to add research perspectives. We have added this sentence further study of the mechanism how NRG4 plays a protective role in patient with diabetic nephropathy would be necessary in the future.

Reviewer 2 Report

Thank you for the opportunity to review this manuscript. I agree that it is a very important topic as DKD is becoming an ever-increasing problem. 

Would suggest few corrections:

Line 42 - should mention insulin resistance as the underlying pathophysiology of T2DM

Line 44 - hyperglycemia and insulin resistance are both responsible for microvascular and microvascular complications

Line 76 - delete metabolic syndrome

General comments: use "people living with diabetes" instead of "diabetic people"

Author Response

Response to Reviewer 2 Comments

Point 1: Line 42 - should mention insulin resistance as the underlying pathophysiology of T2DM

Response 1: We are truly grateful for your valuable comments and suggestions. We have added the underlying pathophysiology of T2DM in introduction.

Point 2: Line 44 - hyperglycemia and insulin resistance are both responsible for microvascular and microvascular complications

Response 2: Thank you very much for the comments. We have added the following sentence in the introduction. Podocyte insulin resistance is a main cause of podocyte injury, playing crucial roles by contributing to albuminuria in early DN.

Reference:Lu J, Chen PP, Zhang JX, Li XQ, Wang GH, Yuan BY, Huang SJ, Liu XQ, Jiang TT, Wang MY, Liu WT, Ruan XZ, Liu BC, Ma KL. GPR43 deficiency protects against podocyte insulin resistance in diabetic nephropathy through the restoration of AMPKα activity. Theranostics. 2021;11(10):4728-4742. doi:10.7150/thno.56598

Point 3: Line 76 - delete metabolic syndrome

Response 3: We are truly grateful for your valuable comments and suggestions. We apologize for this mistake. It is already deleted.

Point 4: General comments: use "people living with diabetes" instead of "diabetic people"

Response 4: Thank you very much for the comments. We have modified this in line 39.